# A Cu/Polypyrrole-Coated Stainless Steel Mesh Membrane Cathode for Highly Efficient Electrocoagulation-Coupling Anti-Fouling Membrane Filtration

Yuna Li [1,†], Zixin Hao [1,†], Jinglong Han [2], Yueyang Sun [1], Mengyao He [1], Yuang Yao [1], Fuhao Yang [1], Meijun Liu [1,*] and Haifeng Zhang [1,*]

1    School of Chemical Engineering, Northeast Electric Power University, Jilin 132012, China
2    School of Civil & Environmental Engineering, Harbin Institute of Technology, Shenzhen 518055, China
*    Correspondence: meijun_liu1@163.com (M.L.); zhfeepu@163.com (H.Z.); Tel.: +86-0432-64806371 (M.L.)
†    The authors contributed to the work equally.

**Abstract:** Membrane filtration fouling has become a significant issue that restricts its wide application. The electrocoagulation (EC) technique combines a variety of synergistic pollutant removal technologies (including flocculation, redox, and air flotation), which can be an ideal pretreatment process for membrane filtration. In this work, a novel $Cu^{2+}$-doped and polypyrrole-coated stainless steel mesh membrane (Cu/PPy–SSM) was prepared by direct current electrodeposition, and it was introduced in an electrocoagulation-membrane reactor (ECMR) to construct an EC–membrane filtration coupling system. The Cu/PPy–SSM was applied as the cathode, while an aluminum plate was used as the anode in the ECMR. The ECMR enabled an excellent humic acid (HA) removal performance and could effectively mitigate the fouling of the Cu/PPy–SSM. Its performance can be attributed to the following: (1) the Cu/PPy–SSM can repel the negatively charged pollutants under the applied electric field; (2) the cathodic hydrogen gas produced on the Cu/PPy–SSM restrains the compacting of the cake layer and delays degradation of membrane flux; and (3) the resultant porous loose structure can perform as a dynamic membrane, which can effectively promote the separation performance of the Cu/PPy–SSM. The resultant ECMR enabled an improved HA removal rate of 92.77%, and the membrane-specific flux could be stabilized at more than 86%. Response surface methodology (RSM) was used to optimize the operation parameters of the ECMR, and the predicted HA removal rate reached 93.01%. Both the experimental results and modelled predictions show that using the Cu/PPy–SSM as a cathode can lead to excellent performance of the ECMR.

**Keywords:** electrocoagulation-membrane reactor (ECMR); membrane fouling; electrocoagulation (EC); polypyrrole (PPy); response surface methodology (RSM)



## 1. Introduction

With the rapid growth of the global population and the acceleration of urbanization, water pollution and the shortage of water resources have become important factors restricting the sustainable and stable development of the social economy [1,2]. Therefore, the development of efficient water treatment and reuse technologies is of great importance for economic and sustainable development. The electrocoagulation (EC) method combines the advantages of coagulation, air flotation, and electrochemistry, and it has been widely used in the field of water treatment [3,4]. In EC, the sacrificial Fe/Al anode produces Fe/Al ions ($Al^{3+}$ and $Fe^{3+}$ (or $Fe^{2+}$)) to coagulate colloids and organics, which can enhance membrane separation and mitigate membrane fouling by improving the cake layer structure. Therefore, using EC as a pretreatment step is a promising method to remove pollutants and mitigate membrane fouling [5–8].

In recent years, the coupling process of electrocoagulation and membrane filtration has attracted more and more attention [9,10]. Membrane filtration is generally located

inside or outside of an EC reactor and uses conventional ultrafiltration (UF) or microfiltration (MF) membranes [11–14]. The filter's cake layer structure is further improved by the combined effect of flocculation and flocculant polarization generated by an applied electric field-induced electrostatic force. In addition, using the conductive membrane as a cathode (membrane cathode), the hydrogen precipitation reaction on the cathode causes the formation of a loose and porous filter cake layer on the membrane surface, which allows the in situ cleaning of the membrane surface [15–18]. Since most pollutants in water are negatively charged, they can be electrostatically repelled by a cathode of the membrane and migrate out of the membrane surface [19–25].

Negatively charged pollutants primarily include natural organic matter (NOM), bacteria, etc. Nearly 50–90% of the NOM in water is humic acid (HA). In order to strengthen the control of organic pollutants in water, the development of conductive membranes (e.g., PPy, DD3R, HFMMMs, ZIF–62, etc.) and the use of electrochemical mechanisms to prevent the pollution of a membrane's surface have become the focus of many studies in recent years [26–32]. To obtain better conductivity of the conducting membranes, conducting polymers with conjugated chains, such as polypyrrole (PPy), have been extensively investigated. PPy is highly conductive and environmentally stable, and it is a heterocyclic conjugated conductive polymeric organic material with monomer molecules which are easy to polymerize by chemical or electrochemical oxidation [33]. Conductive membranes made from PPy have been used in practical applications for wastewater purification, pervaporation, and selective ion separation [34]. From this, it can be predicted that membrane contamination can be further mitigated by using conductive membranes as electrode cathodes under the combined effect of an EC and electric field.

In this study, an electrodeposition membrane Cu/PPy–SSM was prepared as a cathode using a simple electroplating process to construct a novel electrocoagulation membrane reactor (ECMR) with integrated EC and membrane filtration. The range of operating parameters was first selected by single-factor experiments, and then it was determined by BBD–based response surface methodology (RSM), from which the synergistic effect of the EC and membrane filtration on the system's performance was determined. The treatment efficiency and membrane contamination of the Cu/PPy–SSM in the ECMR system were investigated. This study provides a theoretical basis for the construction of a new type of high–efficiency reactor, which is an important guide for achieving the low–cost and high–efficiency removal of pollutants in practical engineering applications compared to electrochemical systems.

## 2. Materials and Methods

### 2.1. Cell Architecture Preparation of the Cu/PPy-SSM

A 400-mesh stainless steel mesh (SSM) (Chuanglei Wire Mesh Products Co., Ltd., Nanyang, China) was used for the experiments. First, the SSM (20 cm × 10 cm) was preliminarily polished with fine sandpaper to remove the surface oxide layer. It was then soaked with alkaline lotion and maintained at a constant temperature of 90 °C for 30 min to remove the oil stain. Subsequently, the SSM was washed in 30 wt% HCl solution for 5 min and then in 10 wt% $H_2SO_4$ solution for 5 min, respectively, to remove the passivation layer on the surface of stainless steel so that the surface had a purified film of relatively high purity and thickness to ensure the integrity and stability of the membrane. Finally, it was subjected to an ultrasound in ultrapure water for 10 min and dried at room temperature.

The electrodeposition preparation method of the PPy–SSM used in this research was optimized from Yang's work. Typically, 0.1 mol/L pyrroles and a 0.1 mol/L sulfuric acid solution are mixed in a ratio of 1:1. To this, $CuSO_4$ was then added at a concentration of 0.05 mol/L to obtain an electrochemical polymerization solution. For the electrodeposition process, a DC (3 A, 35 V Dahua, DH1760, Beijing, China) was used as a power supply while an SSM was used as the anode and a graphite plate with an equal area was used as the cathode. The cathode and anode plates were set in parallel, with a distance of 1.5 cm between them [35]. The SSM electrodeposited at a constant potential of

1.0 V for 60 min [36]. Then, it was taken out and washed with ultrapure water to remove the surface residual pyrrole monomer. The resulting Cu/PPy–SSM was dried at room temperature for future use. For comparison, an electrochemical polymerization solution without CuSO$_4$ was made to prepare the Cu/PPy–SSM, following the same method. All the chemical reagents (analytical grade, purity of $\geq$ 98%) were provided by Sinopharm Chemical Reagent Co., Ltd. (Shanghai, China).

*2.2. ECMR Design*

An ECMR with an effective volume of 3.5 L was constructed with the Cu/PPy–SSM, as shown in Figure 1. An aluminum plate (purity of 99.00%) was used as the anode (20 cm × 10 cm × 2 mm) and the Cu/PPy–SSM (20 cm × 10 cm) as the cathode. The perforated aeration tubes were arranged below the flat membrane module, which provided membrane surface shear. A mechanical stirring device was set on one side of the reactor, and a pressure sensor was used to detect the transmembrane pressure (TMP). The effluent was discharged by a peristaltic pump with artificially simulated micro-polluted water as the feed stream.

For the artificially simulated micro-polluted water, 25 mg HA (Sigma-Aldrich, St. Louis, MO, USA) was dissolved in 1 L of a NaCl solution with a concentration of 0.1 M. Kaolin was subsequently added to adjust the turbidity at 30–40 NTU, while the pH fluctuated between 7.1 and 7.8 (pH probe, STARTER 2100, Shanghai Precision Science Instrument Co., Ltd., Shanghai, China). The HA concentration was determined using an ultraviolet–visible spectrophotometer (U–3010, Hitachi High Technologies Co., Minato-Ku, Japan).

To study the effect origin of the Cu/PPy–SSM in the ECMR, three contrast reactors were constructed (Figure 1). Reactor 1 conducted direct filtering with a Cu/PPy–SSM (Figure 1a), and Reactor 2 individually combined the Cu/PPy–SSM filtration with the EC system (Figure 1b). Reactor 3 was the ECMR we used in the experiments, which coupled filtration with the EC system by using the Cu/PPy–SSM as the cathode and an Al plate as the anode (Figure 1c).

*2.3. Analytical Methods*

The morphologies of the PPy–SSM and the Cu/PPy–SSM were evaluated by a scanning electron microscope (SEM) (JSM7401F, JEDL, Showisland, Tokyo, Japan). The chemical functional groups of the Cu/PPy-SSM were characterized by a Fourier infrared spectroscopy (ATR-FTIR) (Shimadzu IR Prestige-21) infrared spectrometer using Japan and Raman spectra. The electrochemical properties of the Cu/PPy–SSM were investigated on an electrochemical workstation (CHI608E, Shanghai, China). The cyclic voltammetry (CV) between the potential range of −0.3 and 0.7 V (vs. Ag/AgCl) was tested at a scan rate of 10 mV/s. The electron transfer capability of the samples was also studied based on electrochemical impedance spectroscopy (EIS) with an amplitude of 5 mV and a frequency range of 0.01–100,000 Hz.

The membrane flux was determined based on the Darcy equation [37]:

$$J = \frac{\Delta P}{\mu \cdot R_t},$$ (1)

where $J$ represents the membrane flux in L/(m$^2$·h), $\Delta P$ represents the TMP in Pa, $\mu$ represents the kinetic viscosity of the effluent water in Pa·s, and $R_t$ is the total filtration resistance in m$^{-1}$. $R_t = R_m$ when the ultrapure water is applied for the filtration process.

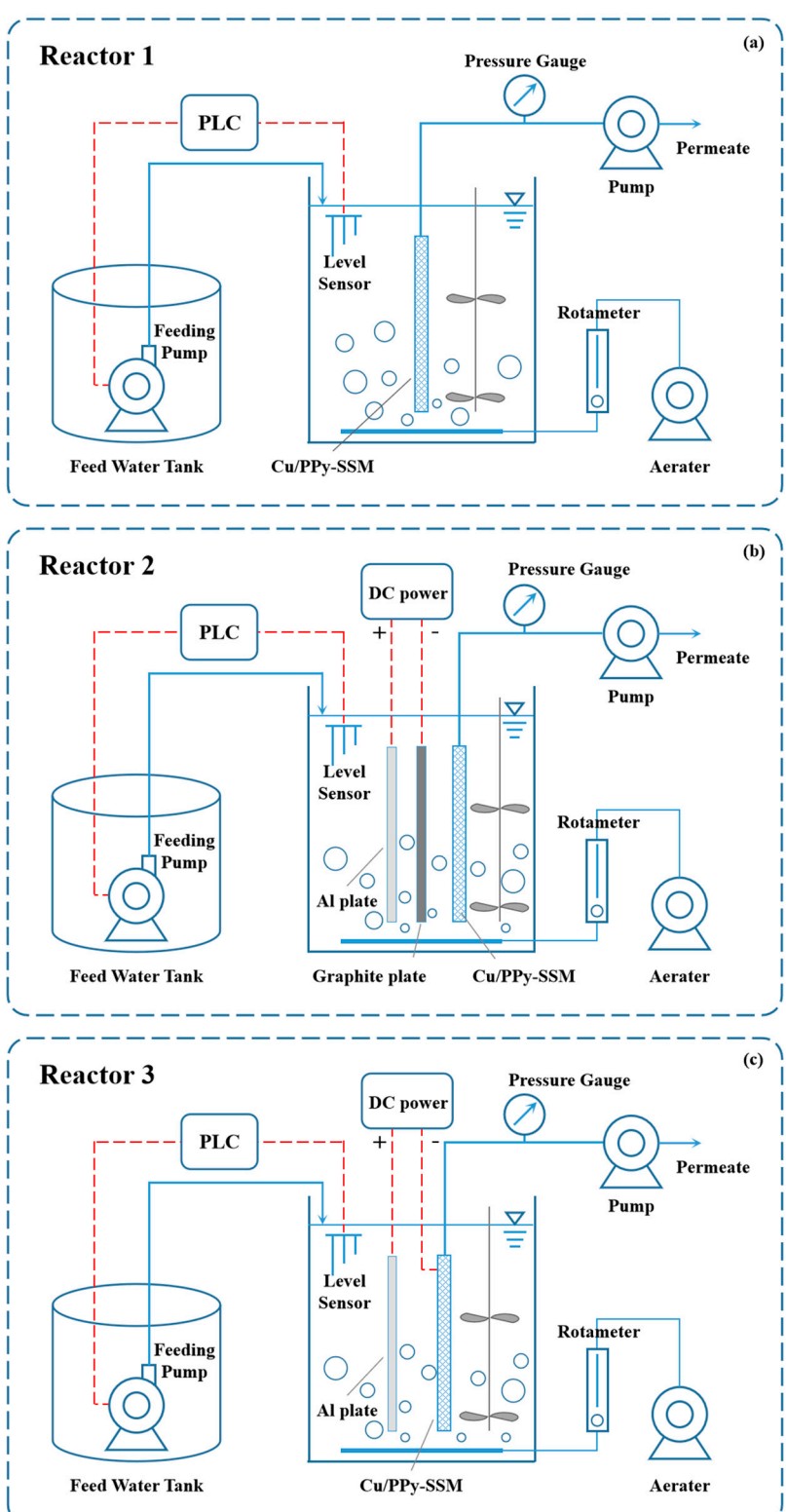

**Figure 1.** (**a**) The Cu/PPy–SSM direct filtration of raw water reactor (Reactor 1). (**b**) The combined Cu/PPy–SSM filtration and EC system reactor, individually (Reactor 2). (**c**) The Cu/PPy–SSM-loaded cathode coupled with the EC reactor (Reactor 3).

### 2.4. Statistical Analysis and Parameters Optimization

Based on the experimental data, the response surface methodology (RSM) was used to compile the statistical data and fit the regression equations. The initial feed water pH

($X_1$), current density ($X_2$), and electrode space ($X_3$) were set as independent variables, and the optimal range of them was determined by single-factor experiments. The optimal operating parameters of the ECMR were obtained following the three-level, three-factor BBD method, and the HA removal rate was used as the response value to establish a quadratic multinomial regression equation. The second-order polynomial model used in the RSM was defined as [38]:

$$Y = A_0 + \sum_{i=1} A_i X_i + \sum_{i=1} A_{ii} X_i^2 + \sum_{i=1}^{n} A_{ij} X_i X_j, \tag{2}$$

where $Y$ is the response variable (HA removal rate); $A_0$, $A_i$, $A_{ii}$, and $A_{ij}$ are the regression coefficients of the constant term, linear term, quadratic term, and interaction term, respectively; and $X_j$ is the independent variable ($i \neq j$). Design Expert software was used to statistically analyze the experimental data and to optimize the reactor operating parameters. Analysis of variance (ANOVA) was used to evaluate the effects of model variables, interactions, and statistical significance. The determination coefficient of ($R^2$) was used to express the fitness of the polynomial model equation, and the F-test was used to test the statistical significance of the regression coefficients at probabilities (P) of 0.001, 0.01, or 0.05. The optimized operating parameters of the ECMR separation reactor were determined by regression analysis and a three–dimensional response surface diagram.

Based on the mathematical model established by the above method, the optimal operating conditions for the HA removal rate were determined and three parallel experimental tests were conducted to investigate the feasibility of the RSM results. The parameters of the respective variables were applied to the three systems with HA as the target pollutant, which investigated the removal of pollutants and membrane contamination of the Cu/PPy–SSM in the EC system.

## 3. Results and Discussions

### 3.1. Characterization

3.1.1. Material Morphology and Microstructure Analysis

Figure 2a–c shows the SEM images of the PPy–SSM, where the membrane pore size was approximately 35 μm, which was slightly lower than the pore size of the original SSM (38 μm). The longitudinal image of the PPy–SSM did not present a large number of polymers, which indicated that there was less polymerization of the pyrrole monomer on the stainless-steel wires. Nevertheless, Figure 2d–f clearly shows that a large amount of polymer was coated on stainless steel wires, and the rough structure of the PPy was clearly visible. This indicated that the electrodeposited Cu/PPy–SSM significantly increased the diameter of the stainless-steel wire, and thus, it effectively reduced the membrane pore size.

The information on the functional groups on the surface of the Cu/PPy–SSM was tested by FT–IR and Raman spectroscopy (Figure 3a,b). As shown in Figure 3a, in the FT-IR spectra of the Cu/PPy–SSM, the characteristic peaks of PPy clearly appear, among which 3295 cm$^{-1}$ corresponds to the stretched vibration absorption peak of N-H and the absorption peaks at 2922 cm$^{-1}$ and 2852 cm$^{-1}$ are related to the stretching vibration absorption peaks of -CH$_3$ or -CH$_2$-, respectively, while 1612, 1416, and 1248 cm$^{-1}$ are raised from the C=C, C-C, and C-N characteristic absorption peaks, respectively. The above characteristic absorption peaks indicate that the PPy had been escapist on the surface of the SSM, and thus, they became an important component of the membrane.

The characteristic peaks of the Raman spectrum at 929 cm$^{-1}$, 964 cm$^{-1}$, and 1046 cm$^{-1}$ can be ascribed to the ring deformation vibration of the C-H bond, while the peak at 1249 cm$^{-1}$ can be ascribed to the in-plane bending vibration of the C-H bond (Figure 3b). The characteristic peaks at 1325 cm$^{-1}$, 1387 cm$^{-1}$, and 1570 cm$^{-1}$ can be ascribed to the stretching vibrations of the C-N, C-C, and C=C bonds, respectively. The results of the

Raman spectrum further confirmed the successful coating of PPy on the surface of the SSM, which created the improved electrochemical properties of the Cu/PPy–SSM.

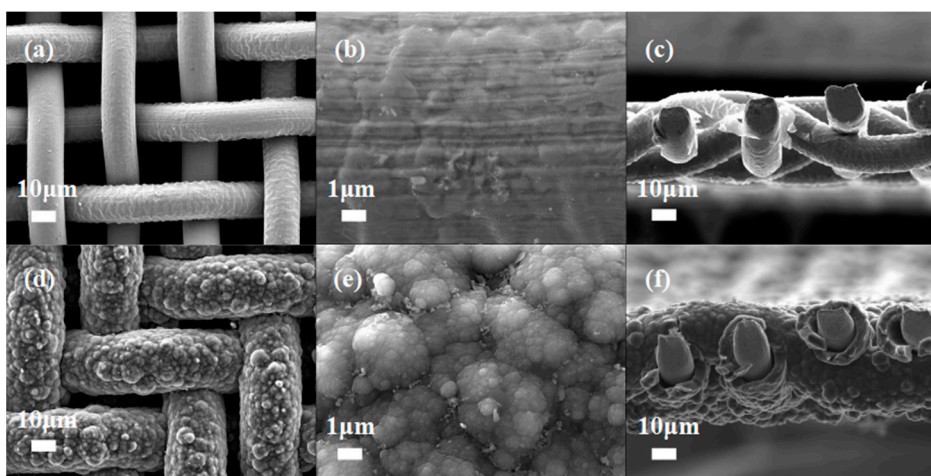

**Figure 2.** SEM images of the plane (**a**,**b**) and section (**c**) of the PPy–SSM at 60 min of electrodeposition. SEM images of the plane (**d**,**e**) and section (**f**) of the Cu/PPy–SSM at 60 min of electrodeposition.

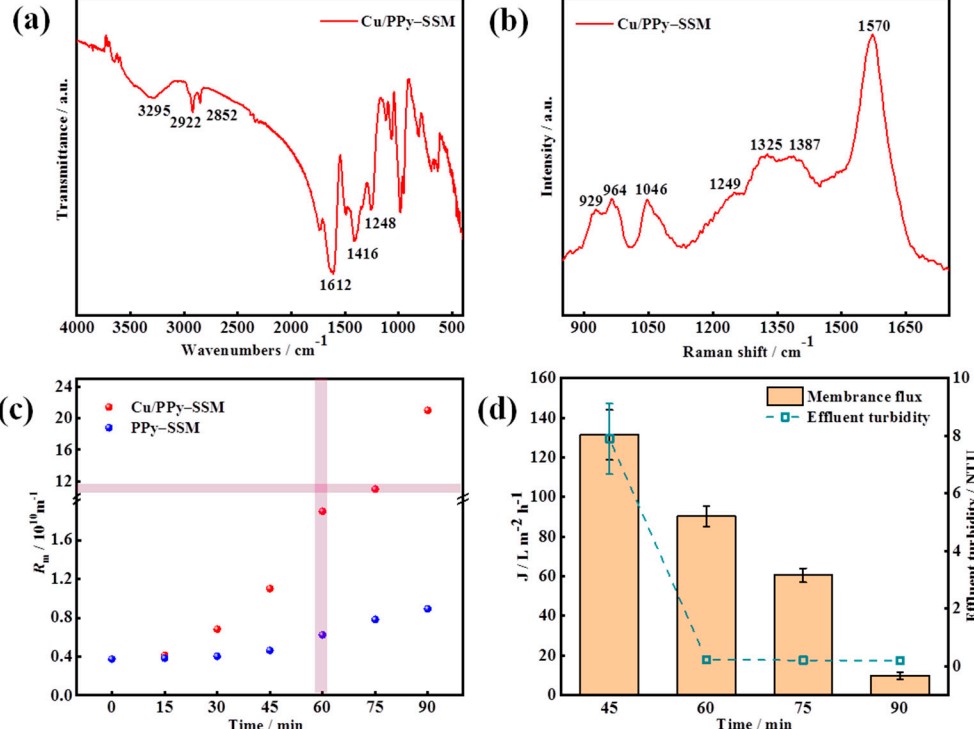

**Figure 3.** FT–IR spectra (**a**) and Raman spectra (**b**) of the Cu/PPy–SSM. (**c**) Variation in the $R_m$ of the Cu/PPy–SSM with electrodeposition time. (**d**) Effect of the electrodeposition time of the Cu/PPy–SSM on the stable membrane flux and effluent turbidity.

### 3.1.2. Effect of Electrodeposition Time on Membrane Properties

To optimize the synthesis parameters, the performance of the Cu/PPy–SSM under different electrodeposition times has been studied. As one of the most important performance indexes, the intrinsic resistance ($R_m$) reflects the porosity and retention capacity of a separation membrane. As shown in Figure 3c, the $R_m$ of the PPy–SSM slowly increased from $0.40 \times 10^{10}$ m$^{-1}$ (30 min) to $0.89 \times 10^{10}$ m$^{-1}$ (90 min) with the prolonged electrodeposition time. Nevertheless, the $R_m$ of the Cu/PPy–SSM increased rapidly 31 times from

$0.68 \times 10^{10}$ m$^{-1}$ (30 min) to $21 \times 10^{10}$ m$^{-1}$ (90 min) with the prolonged electrodeposition time. The presence of Cu$^{2+}$ in the electrochemical polymerization solution significantly promoted the electrodeposition process, which may be related to the coordination effect between Cu$^{2+}$ and the pyrrole monomer [39]. It is worth noting that the R$_m$ of the Cu/PPy–SSM increased sharply from $1.90 \times 10^{10}$ m$^{-1}$ to $11 \times 10^{10}$ m$^{-1}$ within the period of 60–75 min. This indicated that the electrodeposition process appeared to have thickened the wire diameter of the SSM in the first 60 min, while part of the mesh closed and significantly increased the R$_m$ of the Cu/PPy–SSM during the remaining electrodeposition time.

The membrane flux and effluent turbidity of the Cu/PPy–SSM was further studied under different electrodeposition times (Figure 3d). The membrane flux decreased obviously from 131.32 L/m$^2$·h (45 min) to 9.78 L/m$^2$·h (90 min) with the prolonged electrodeposition time under 30 kPa constant filtration pressure, which was consistent with the trend of the $R_m$. The gradual decrease in the SSM pore size with the electrodeposition time led to a rapid decrease in the membrane flux, which also increased the retention capacity of the Cu/PPy–SSM and effectively improved the effluent water quality. The effluent turbidity decreased sharply from 7.89 NTU (45 min) to 0.19–0.22 NTU (60–90 min), which was related to the formation of the dynamic membrane under the appropriate pore size of the Cu/PPy–SSM. Based on the above results, the electrodeposition time of the Cu/PPy–SSM selected for the research that followed was 60 min.

When the Cu/PPy–SSM was applied as the cathode in the ECMR, it was charged negatively under the electric field. It could repel HA with the same charge away from the membrane, which would effectively reduce membrane pollution. The membrane–specific flux of the Cu/PPy–SSM with and without voltage was further investigated with the operation times shown in Figure 4. The membrane–specific flux gradually decreased with operation time when no voltage was applied, which indicated that membrane fouling occurred along with membrane filtration. However, the specific flux of the Cu/PPy-SSM performed continued fluctuations instead of a sustained decline under a voltage of 2 V, and it remained at 83% after 180 minutes of operation. This revealed the anti–pollution mechanism of the Cu/PPy–SSM by being negatively charged under an electric field.

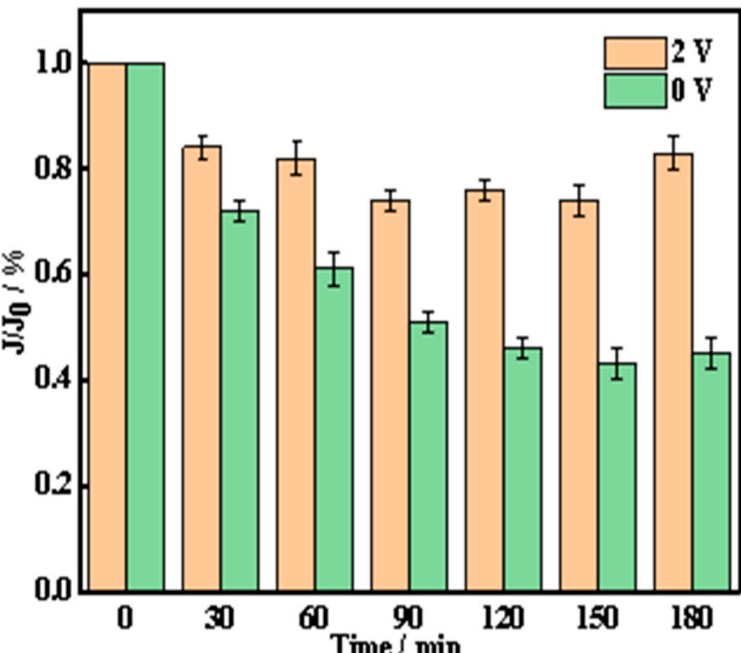

**Figure 4.** Effect of presence or absence of voltage on the specific flux of the Cu/PPy–SSM.

### 3.1.3. Electrochemical Characterization

The electrochemical CV curves of the Cu/PPy–SSM and SSM at the 11th–15th cycles were tested to investigate their electrochemical performance (Figure 5a). The current response of the Cu/PPy–SSM was significantly higher than that of the SSM, which was related to the higher electrochemical activity of the Cu/PPy–SSM. No obvious redox peaks were shown on the Cu/PPy–SSM, and the CV curves greatly overlapped in each cycle. The electron transfer abilities of the Cu/PPy–SSM and SSM were further investigated by EIS (Figure 5b). Compared with the SSM, the Cu/PPy–SSM performed a capacitive resistance arc with a smaller diameter at the high–frequency part, which corresponded to the faster electron transfer rate at the electrode surface. The Cu/PPy–SSM showed a straight line (charge saturation) at the middle frequency part and charge saturation diffusion at the low–frequency part. The electron transfer resistance of the Cu/PPy–SSM was smaller than that of the SSM, indicating the higher electron transfer rate of the Cu/PPy–SSM. The dense, conductive PPy coated on the surface of the SSM effectively improved the conductivity and stability of the Cu/PPy–SSM.

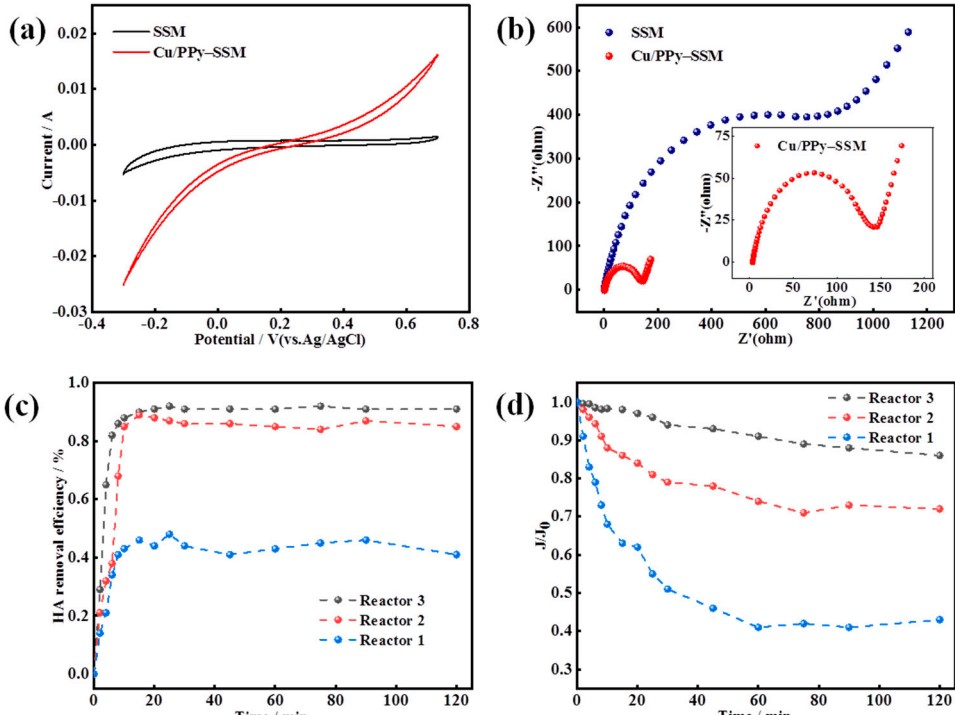

**Figure 5.** Cyclic voltammetry (**a**) and EIS (**b**) tests of the SSM and Cu/PPy–SSM (the inset is a magnification of the Cu/PPy–SSM EIS test), and HA removal rates (**c**) and membrane–specific fluxes (**d**) for the different reactor units.

### 3.2. Performance of the ECMR

The performance of the three reactors was analyzed to evaluate the effect of the Cu/PPy–SSM in the ECMR. As shown in Figure 5c, all three reactors achieved a stable state within approximately 10 min, which corresponded to the dynamic membrane formation time. The HA removal rate by reactor 1 was approximately 40–50%, which indicated that the Cu/PPy–SSM filtration alone was not satisfactory for organic pollutants removal. In reactor 2, the particle size of the pollutants was effectively increased by the compressed double layer, adsorption electro–neutralization, and electrophoresis and oxidation reaction mechanisms of the EC system. The effluent HA removal rate of the improved Cu/PPy–SSM filtration process reached 85.26%, which revealed the advantage of the membrane filtration and EC coupling system. The HA removal rate of reactor 3 was further improved to 88.31%, which revealed that the introduction of the Cu/PPy–SSM effectively improved

the pollutant treatment performance of the ECMR. The HA removal rate was improved by approximately 16%, which was relative to the electric flocculation processes in other studies [40].

In order to investigate the anti–fouling effect of the Cu/PPy-SSM in the ECMR system, the membrane–specific flux with a running time of the three reactors was tested, as shown in Figure 5d. The Cu/PPy–SSM specific flux of reactor 1 achieved stability after 60 min, and it decayed nearly 60% within the operation time. This decay originated from a large number of pollutants gathered in the membrane pores during membrane filtration, which revealed serious membrane fouling in the individual Cu/PPy–SSMs. The decay time of the membrane-specific flux for reactor 2 was approximately 30 min, and the stabilized specific flux of the Cu/PPy–SSM remained above 72%. The combination of EC with membrane filtration effectively mitigated membrane fouling. Compared with the above two reactors, the negatively charged Cu/PPy–SSM significantly reduced the membrane fouling rate. The specific flux of the Cu/PPy–SSM in the ECMR did not show significant decay with operation time, and it showed a slow linear decrease that remained at more than 86% of the membrane–specific flux. The superiority of the Cu/PPy–SSM as the cathode of the ECMR can be attributed to: (1) the negatively charged Cu/PPy–SSM's ability to effectively repel the pollutants on the membrane surface under an electric field, and (2) the hydrogen gas generated by the cathode reaction, which acted as a scrubbing agent and which resulted in a loose cake layer on the surface of the Cu/PPy–SSM (Figure 6). The above results revealed the advantage of introducing the Cu/PPy–SSM to the ECMR. It can not only promote pollutant removal efficiency, but it can also reduce the degradation caused by membrane fouling.

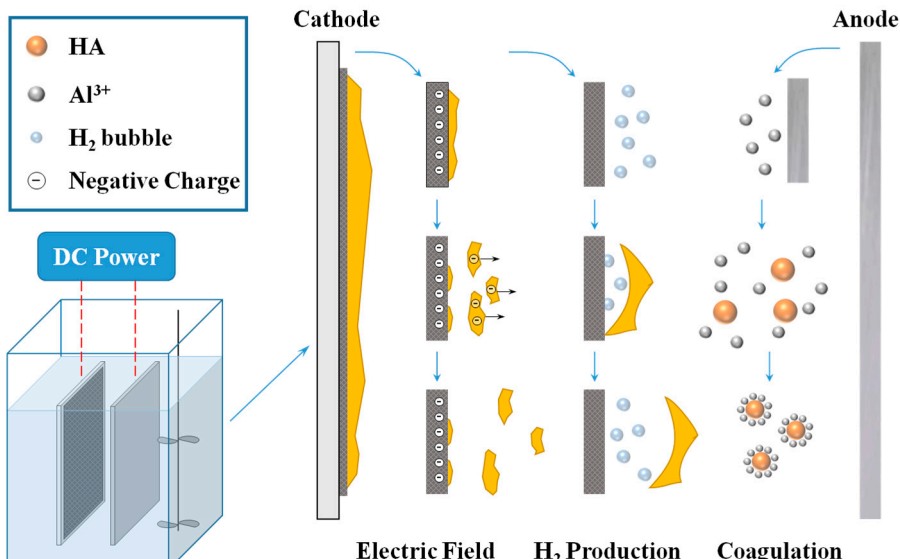

**Figure 6.** The schematic diagram for the mechanism of mitigating membrane cathode fouling.

### 3.3. Single-Factor Experiments

To further optimize the operating parameters of the ECMR, we carried out single-factor experiments to select the range of optimization parameters. As shown in Figure 7, the HA removal rate was set as the main index, while the pH, electrode space, and current density were selected as the independent variables.

### 3.3.1. Effect of pH on the HA Removal Efficiency

The current density was fixed at 2 mA/cm$^2$ and the electrode space was fixed at 2.5 cm, while the pH was adjusted to 4, 5, 6, 7, 8, 9, and 10 to investigate the effect of these variables on the HA removal rate of the ECMR. As shown in Figure 7a, with the increase in pH, the HA removal rate showed a tendency to increase at a pH of <7 and then decreased at a pH of >7. The influence of pH on HA removal was mainly determined by the state of the

Al$^{3+}$ hydrolysate [41]. The Al coagulant formed by EC under acidic conditions primarily consisted of a comonomer, which became medium aggregates with good adsorption and aggregation capacities under a neutral pH [42]. The hydrolysis products of Al under alkaline conditions were negative ions, such as Al(OH)$_4^-$, which reduced the pollutant aggregation ability of the electrostatic repulsivity with HA. Therefore, the optimized pH range was set as 6–8, under which the HA removal rate exceeded 86%.

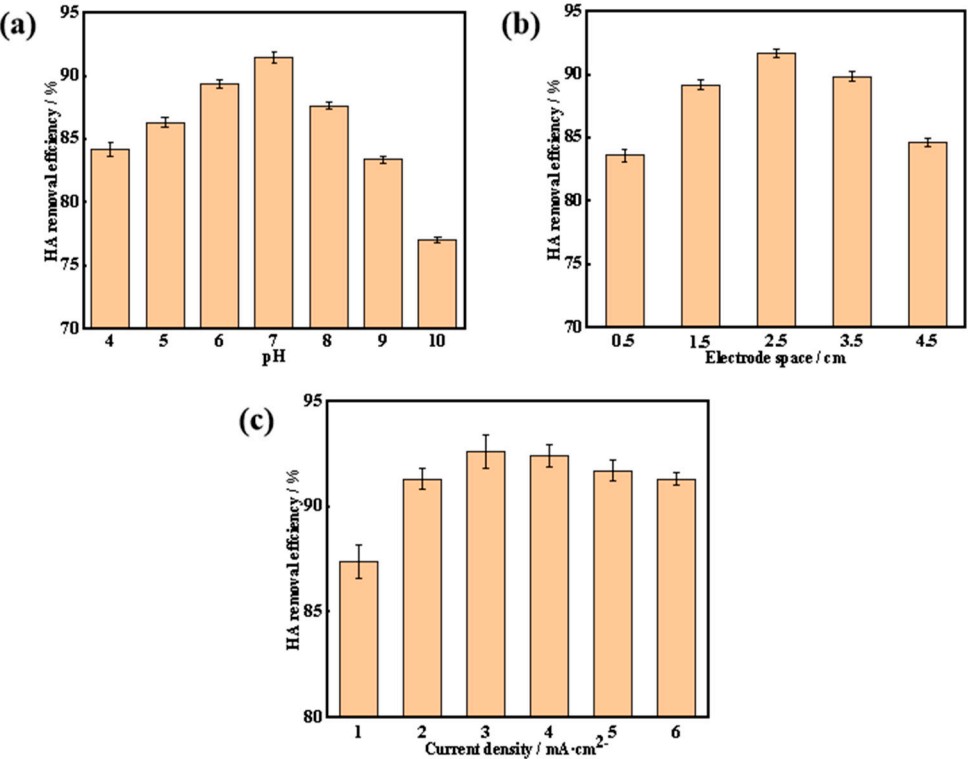

**Figure 7.** Effect of pH (**a**), electrode space (**b**), and current density (**c**) on HA removal rate.

### 3.3.2. Effect of Electrode Space on the HA Removal Efficiency

Electrode space was another important factor affecting the HA removal rate. As shown in Figure 7b, the removal rate of HA gradually increased and peaked at 2.5 cm, and then it decreased with the increase in the plate spacing. The electrode space determined the current density of the ECMR under constant voltage conditions. The diffusion distance for Al ions greatly increased under larger electrode space, which affected the HA removal rate. The removal rate of HA was higher than 85% under the electrode space range of 1.5–3.5 cm, and it was selected for the following optimization.

### 3.3.3. Effect of Current Density on the HA Removal Efficiency

Current density directly affects the dissolution rate of Al from an anode plate during the operation under a constant voltage operation mode. Figure 7c shows the effect of the current density on the HA removal efficiency, which gradually increased and then leveled off with increasing current density. It achieved a peak value at 3 mA/cm$^2$, followed by a small decrease in tendency. The reasons for the decrease in the HA removal rate under high current density may be: (1) The charge on the colloidal surface revealed and resulted in the "re-mixing" phenomenon under the high concentration of aluminum [43]. (2) The increased hydrogen product rate at high current densities affected the formation of the cake layer on the Cu/PPy–SSM, resulting in reduced retention performance. Taking the energy consumption into consideration, the current density was optimal between 1 and 5 mA/cm$^2$.

### 3.4. Optimization of the ECMR Operating Parameters by RSM

Based on the single-factor experimental results, the level of the three factors was coded as shown in Table 1. The BBD included 17 response surface analysis experiments, of which 12 were factor points and 5 were central points.

**Table 1.** Response surface factor horizontal coding.

| Factor | Code | Coding Level | | |
|---|---|---|---|---|
| | | **−1** | **0** | **1** |
| pH | A | 6 | 7 | 8 |
| Plate spacing (cm) | B | 1.5 | 2.5 | 3.5 |
| Current density (mA/cm$^2$) | C | 1 | 3 | 5 |

The response values in Table 2 were fitted by the Design-Expert software, and the second-order polynomial regression equations between pH ($X_1$), electrode space ($X_2$), current density ($X_3$), and HA removal rate ($Y$) were obtained by using Equation (3):

$$Y = 92.55 - 1.62X_1 + 0.25X_2 + 1.73X_3 - 0.095X_1X_2 + 0.1X_1X_3 - 0.21X_2X_3 - 4.85X_1{}^2 - 1.91X_2{}^2 - 2.32X_3{}^2 \ (R^2 = 0.9944). \quad (3)$$

The significance of the second-order multinomial regression model was tested using the three factors and the HA removal rate (Table 3). The results of the analysis of variance (ANOVA) showed that the linear correlation of both $X_1$ and $X_3$ on $Y$ was extremely significant ($p < 0.001$), while the linear effect between $X_2$ and $Y$ was not significant ($p > 0.05$). In the second-order terms, $X_1{}^2$, $X_2{}^2$, and $X_3{}^2$ each had a significant influence on $Y$, while $X_1X_2$, $X_1X_3$, and $X_2X_3$ each had a insignificant influence on $Y$ [44]. The F-value of the model was 139.25, the $p$-value was <0.0001, and the $p$ lack-of-fit was 0.1638 > 0.05. The results indicated that the established model was feasible and the relationship between the three factors and the HA removal rate was reasonable.

**Table 2.** The test results of the response surface methodology.

| Number | Coding Level | | | Removal of HA Rides (%) |
|---|---|---|---|---|
| | $X_1$ | $X_2$ | $X_3$ | |
| 1 | 0 | 0 | 0 | 92.67 |
| 2 | 1 | −1 | 0 | 84.05 |
| 3 | 0 | 1 | −1 | 86.81 |
| 4 | −1 | 0 | 1 | 88.31 |
| 5 | 0 | 0 | 0 | 92.22 |
| 6 | −1 | 0 | −1 | 85.64 |
| 7 | 0 | 0 | 0 | 92.27 |
| 8 | 0 | 0 | 0 | 92.89 |
| 9 | 0 | −1 | −1 | 85.76 |
| 10 | −1 | −1 | 0 | 87.15 |
| 11 | 1 | 0 | 1 | 85.31 |
| 12 | 0 | −1 | 1 | 90.25 |
| 13 | 1 | 1 | 0 | 84.23 |
| 14 | 1 | 0 | −1 | 82.24 |
| 15 | 0 | 0 | 0 | 92.71 |
| 16 | −1 | 1 | 0 | 87.71 |
| 17 | 0 | 1 | 1 | 90.45 |

**Table 3.** Results of ANOVA.

| Source | Sum of Squares | df | Mean Square | F Value | *p*-Value Probability > f |
|---|---|---|---|---|---|
| Model | 195.77 | 9 | 21.75 | 139.25 | <0.0001 |
| $X_1$ | 21.06 | 1 | 21.06 | 134.82 | <0.0001 |

**Table 3.** *Cont.*

| Source | Sum of Squares | df | Mean Square | F Value | p-Value Probability > f |
|---|---|---|---|---|---|
| $X_2$ | 0.50 | 1 | 0.50 | 3.17 | 0.1183 |
| $X_3$ | 24.05 | 1 | 24.05 | 153.94 | <0.0001 |
| $X_1X_2$ | 0.036 | 1 | 0.036 | 0.23 | 0.6454 |
| $X_1X_3$ | 0.040 | 1 | 0.040 | 0.26 | 0.6284 |
| $X_2X_3$ | 0.18 | 1 | 0.18 | 1.16 | 0.3179 |
| $X_1^2$ | 99.24 | 1 | 99.24 | 635.28 | <0.0001 |
| $X_2^2$ | 15.40 | 1 | 15.40 | 98.57 | <0.0001 |
| $X_3^2$ | 22.71 | 1 | 22.71 | 145.36 | <0.0001 |
| Residual | 1.09 | 7 | 0.16 | | |
| Lack of Fit | 0.75 | 3 | 0.25 | 2.92 | 0.1638 |
| Pure Error | 0.34 | 4 | 0.086 | | |
| Corrected Total | 196.86 | 16 | | | |

Note: $p < 0.001$ is extremely significant, $p < 0.01$ is highly significant, and $p < 0.05$ is significant.

The reliability of the regression equation was analyzed, as shown in Table 4. The model correlation $R^2$ was 0.9944, which indicated the acceptance of the experimental model. The $R^2_{Adj}$-$R^2_{Pred}$ was 0.051 < 0.2, and the coefficient of variation was 0.45% < 10%, which indicated the high reliability and accuracy of the model. The signal-to-noise ratio (SNR) of the experiment reached 35.078 and was much higher than the standard value (4.0). In general, the fit and feasibility of the regression model in each independent variable were at the significance level, indicating that the second-order model could be used for the optimization of the operational parameters in the study of applying the Cu/PPy–SSM for HA removal in the ECMR.

**Table 4.** Reliability analysis of regression equation.

| Item | c.v.% | R-Squared | Adjusted R-Squared | Predetermined R-Squared | Adequate Precision |
|---|---|---|---|---|---|
| Value | 0.45 | 0.9944 | 0.9873 | 0.9363 | 35.078 |

Three-dimensional response surface and contour maps can reflect the relationship between the response value (HA removal rate) and the independent variables (pH, electrode space, and current density), as well as the interaction between the test variables. The red location corresponds to the higher HA removal rate, which can be selected as the target value of the optimization range. The shape and steep tendency of the 3D response surface correspond to the degree of significance of the factor on the response value. The results of applying different pH levels, electrode spaces, and current densities are shown in Figure 8.

As shown in Figure 8a, the HA removal rate increased with the increase in current density at its lower range, and it achieved the best value at the current density of 4.5–5.5 mA/cm$^2$. The value of current density was positively related to the content of the Al$^{3+}$ dissolved from the anode, according to the Faraday effect, and it was directly related to the HA removal effect of EC. The further increase in current density (>4 mA/cm$^2$) did not result in the promotion of the HA removal rate, which may have been related to the re-stabilization of the colloids by the excessive addition of coagulant [45]. Furthermore, the larger current density would result in the passivation of the electrode, and a large amount of H$_2$ produced on the cathode would also affect the formation of the dynamic membrane. It is worth noting that the contour map was close to the circle (P$_{X_2X_3}$ > 0.05), indicating that the interaction between the electrode space and current density was not significant. The highest HA removal rate of the response surface was in the range of 2–3 cm, indicating that there was a maximum response value in this range (Figure 8b). However, the electrode space did not have a significant effect on the HA removal rate, which was related to the poor hydraulic flow under the narrow electrode space and the weakening of the mixing of

$Al^{3+}$ in the solution. The further increase in electrode space to above 3 cm intensified the concentration polarization, which affected the release of $Al^{3+}$ and reduced the HA removal performance of the EC system [46]. The effect of pH was similar to that of the electrode space, but its effect on the HA removal rate was more significant. The response surface and contour map in Figure 8c illustrate that the interaction between the pH and the current density was significant for the HA removal rate.

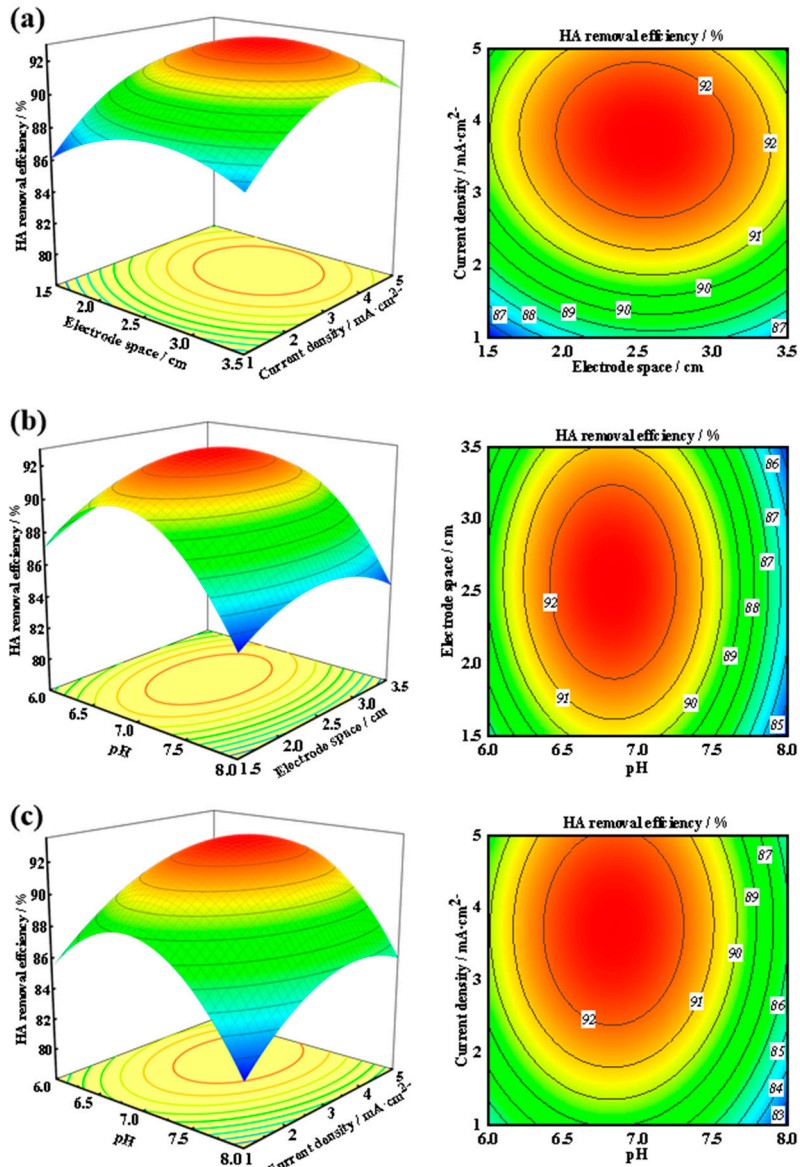

**Figure 8.** Response surface and contour map of the effects of pH ($X_1$), electrode space ($X_2$), and current density ($X_3$) on HA removal. (**a**) HA removal rate with response surface and contour map of $X_2$ and $X_3$. (**b**) HA removal rate with response surface and contour map of $X_1$ and $X_2$. (**c**) HA removal rate with response surface and contour map of $X_1$ and $X_3$.

### 3.5. Consistency Analysis between Model and Optimal Operating Conditions

As shown in Figure 9a, it can be predicted that the maximum response value of the HA removal rate is 93.01% under the minimum energy consumption, based on the model established by the RSM. The optimum operating conditions of the ECMR are as follows: feed water pH of 6.84, electrode space of 2.55 cm, and current density of 3.74 mA/cm$^2$. In order to verify the feasibility of the RSM, the prediction parameters were used to test the ECMR. Considering the actual operation, the optimum conditions of EC were revised

as follows: the pH value of the feed water was 6.8, the electrode space was 2.5 cm, and the current density was 3.7 mA/cm². Figure 9a shows the statistical distribution of the HA removal rate in three parallel experiments. The average HA removal rate in the three parallel experiments was 92.77%, which was close to the theoretical prediction value of 93.01%, verifying the reliability of the model. After operation, the Cu/PPy–SSM was cleaned and its electrochemical performance was investigated. The electrochemical CV and EIS curves of the Cu/PPy–SSM before and after operation were tested. As shown in Figure 9b,c, the electrochemical performance of the membrane had little difference before and after operation, indicating that the membrane could be recycled in the electrochemical system.

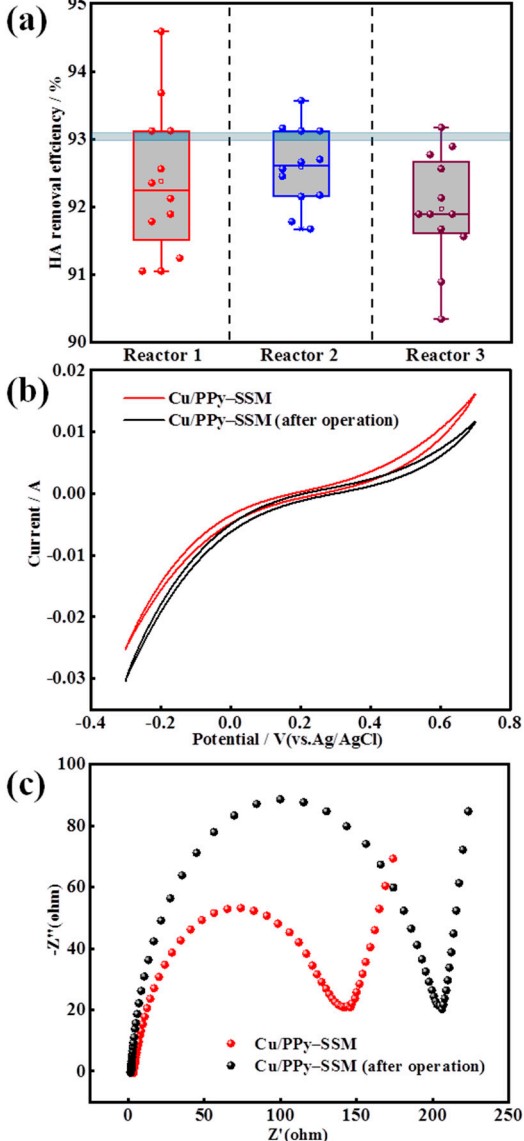

**Figure 9.** (**a**) Experimental verification of the results of the response surface model. Cyclic voltammetry (**b**) and EIS (**c**) tests of the Cu/PPy–SSM and Cu/PPy–SSM (after operation).

## 4. Conclusions

In summary, this research has successfully prepared a novel Cu/PPy-SSM using a simple electrodeposition method, and the constructed ECMR performs well in the application of pollutant removal. PPy was loaded on the stainless-steel net with the acceleration of Cu²⁺, which effectively reduced the pore size of the SSM. The resulting Cu/PPy-SSM performed higher levels of electron transfer and had a better electrochemical stability ability.

A comparison of the three reactors confirmed the advantage of the EC-membrane filtration coupling system, and its HA removal rate and membrane-specific flux were 92.77% and 86%, respectively. The membrane fouling rate of the Cu/PPy-SSM was significantly reduced in the ECMR by the use of electrostatic repulsion and the hydrogen bubble scrubbing of the cathode under the electric field. The optimized HA removal rate by the RSM was 93.01%, which was consistent with the experimental results. The reported ECMR is an important guide for achieving cost-effective and efficient pollutant removal in practical engineering applications, and it has the potential to achieve a better anti-fouling performance in more compact reactors.

**Author Contributions:** Writing—original draft, Data curation, Formal analysis, Y.L. and Z.H.; Writing—review & editing, J.H.; Formal analysis, Y.S.; Investigation, M.H.; Date curation, Y.Y. and F.Y.; Conceptualization, writing—review & editing, M.L.; Supervision, Project administration, H.Z. All authors have read and agreed to the published version of the manuscript.

**Funding:** This research was funded by the National Natural Science Foundation of China (No. 52100075 and 52070035), the Jilin Province Scientific and the Technological Planning Project of China (No. 20220203009SF and 20200403001SF), and the Key Research and Development Project of Shandong Province (No. 2020CXGC011202).

**Institutional Review Board Statement:** Not applicable.

**Informed Consent Statement:** Not applicable.

**Data Availability Statement:** The date presented in this study are available in the article itself.

**Conflicts of Interest:** The authors declare no conflict of interest.

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
