# Peer review of "A Cu/Polypyrrole-Coated Stainless Steel Mesh Membrane Cathode for Highly Efficient Electrocoagulation-Coupling Anti-Fouling Membrane Filtration"

_sustainability, doi:10.3390/su15021107_

Round 1

Reviewer 1 Report

In this manuscript, an electrocoagulation-membrane reactor with an aluminum plate as the anode and a Cu/PPy-SSM as cathode was constructed. The coupling system was found to remove pollutant and relieve membrane fouling effectively. The work has something new, and the manuscript is carefully organized and written. The paper deserves publication. The minor revision is needed as follow before it to be published in Sustainability.

1.     In the abstract section, the authors propose the preparation of a novel type of Cu/PPy-SSM membrane, but the full name did not mention the Cu2+. The full name for “HA” should be given when it first appeared.

2.     In section 2.1, why did the pretreatment process for stainless steel mesh used two kinds of acid?

3.     More detailed chemical information and treatment method need to be provided in Materials and Methods section.

4.     Why did the authors choose HA as the target pollutant instead of other pollutant?

5.     It would be more attractive if the authors can highlight the impact of the current research in industry application.

Reviewer 2 Report

The present manuscript reported a preparation method of a novel polypyrrole-coated stainless steel mesh membrane (Cu/PPy-SSM) and applied it to construct an EC-membrane filtration coupling system. The results indicated improved HA removal performance of coupling system, and it can also effectively mitigate the membrane fouling issue. The novelty of the manuscript is obvious, and the system is new and interesting. Their conclusion was supported by the experimental results appropriately. Therefore, I recommend the manuscript for the publication in Sustainability after minor revision.

1.       In the introduction part, it is suggested that the advantages of EC-membrane should be further illustrated, compared with some other researches reported on membrane modification to control the membrane fouling, e.g. Enhancing the permeability, anti-biofouling performance and long-term stability of TFC nanofiltration membrane by imidazole-modified carboxylated graphene oxide/polyethersulfone substrate, Journal of Membrane Science, 2022,664 or Designing the anti-biofouling surface of an ultrafiltration membrane with a novel zwitterionic poly(aryl ether oxadiazole) containing benzimidazole, Applied Surface Science, 2023, 609, et al.

2.       In the first paragraph of “Introduction” section, what does it mean by “remembrance fouling”?

3.       In the last line of section 2.1, it should be Cu/PPy-SSM rather than PPy-SSM, right?

4.       The determination method for the concentration of target pollutant (HA) needs to be included in the “Materials and Methods” section.

5.       In equation 2, there is an extra “+” between the second term and the third term.

6.       The type size of the inset in Figure 5b need to be adjusted, and the statement of it also need to be included in the Figure caption.

7.       The Al(OH)4- in part 3.3.1 should be adjusted to Al(OH)4-.

Reviewer 3 Report

1. Author need to change the title of the manuscript. It is not reflecting the content and novelty of work

2. Author did not describe the results in abstract

3. Keywords are also not reflecting the content of work

4. The introduction must be reduced to one and a half pages.

5. The authors should describe the importance of their research more clearly. The references cited lack articles on contaminants from last year. Try to concise it in one and half page. Last section must be gist of study.

 6. Why and how the said parameters were selected for this work. More specific details are needed to be added with use of latest reference.

7. Authors also can write about the impact of current work on the future research and industry

8. Highlight the problem statement of the current work in the abstract and conclusion

10.  Check for grammatical errors and typos and improve the literature such as.

https://doi.org/10.1016/j.jngse.2016.01.015

https://doi.org/10.1016/j.seppur.2018.12.086

https://doi.org/10.1016/j.chemosphere.2021.133006

11. Conclusion section needs to be improved as well. Impact on industry is missing as well.

Reviewer 4 Report

In the current manuscript, the direct current electrodeposition was utilized to prepare polypyrrole-coated stainless steel mesh membrane (Cu/PPy-SSM). The Cu/PPy-SSM membrane was used as a cathode in an electrocoagulation cell with an aluminum to mitigate humic acid from water. The results demonstrated that the Cu/PPy-SSM can effectively remove the humic acid as well as mitigate fouling. The manuscript is well structured and fit the scope of the Sustainability journal. However, there are some keys of improvement before considered for publication.

·         More significant data, numbers have to be mentioned in the abstract.

·         The authors have done good introduction, however, before the beginning of the last paragraph, show the literature of using the same polymer, some examples in the literature in the topic. After that, it is very important to show the research gap and how is your work contribute into the literature. Research gap and literature must be added to the introduction. What are the main differences from other studies such as Reference 28, “”https://www.sciencedirect.com/science/article/pii/S0376738820315118”.

·         Material and methods: for each material utilized in the study, you have to provide the manufacturer, purity and country of manufacturer in a systematic way.

·         Similarly, for equipment, the SEM, which model, manufacturer, country. Please, do it for all equipment and in a systematic way?

·         Any characterization of the used cathode, I mean after humic acid removal.

·         The authors have to compare their outcomes with previously reported data in the field either in a separate section or where appropriate. The efficiency in humic acid removal

·         Have the authors considered long term experiments and reusability/cycling of the prepared cathode.

·         Some typos, mistakes, review the article and improve the language of the manuscript, example: (In the abstract can performs, to can perform) (The first sentence of the introduction” has become one of the greatest challenge”, what is it?.

·          It is recommended to show the mechanism of humic acid removal in a separate section with related figure. I mean the repulsions, effect of pH and if there is any chemical mechanism based on characterization.

·         Why the study limited to humic acid removal, sodium alginate and bovine serum albumin might enrich the manuscript if possible.

·         The following references might be included:

https://www.sciencedirect.com/science/article/pii/S0045653522039182

https://www.mdpi.com/2077-0375/12/11/1143

Round 2

Reviewer 4 Report

The authors made significant improvement in the paper, the paper can be accepted in the current form.